# Aetiology of Vulvovaginal Candidiasis in Ecuador and In Vitro Antifungal Activity Against *Candida* Vaginal Isolates

**DOI:** 10.3390/jof11100742

**Published:** 2025-10-16

**Authors:** Celia Bowen, Cristina Marcos-Arias, Carmen Checa, María Eugenia Castellanos, Katherine Miranda-Cadena, Elena Eraso, Guillermo Quindós

**Affiliations:** 1Department of Immunology, Microbiology and Parasitology, Faculty of Medicine and Nursing, University of the Basque Country (UPV/EHU), 48940 Leioa, Bizkaia, Spain; cabowen@puce.edu.ec (C.B.); cristina.marcos@ehu.eus (C.M.-A.); katherine.miranda@ehu.eus (K.M.-C.); elena.eraso@ehu.eus (E.E.); 2Medicine Program, Faculty of Health and Wellbeing, Pontificia Universidad Católica del Ecuador (PUCE), Quito 170525, Ecuador; karmencheca@gmail.com (C.C.); mecbginecologia@gmail.com (M.E.C.); 3Instituto Biosanitario Biobizkaia, 48903 Barakaldo, Bizkaia, Spain

**Keywords:** vulvovaginal candidiasis, *Candida*, Ecuador, antifungal susceptibility, amphotericin B, clotrimazole, fluconazole, itraconazole, miconazole, nystatin

## Abstract

The epidemiology of vulvovaginal candidiasis (VVC) in Ecuador remains poorly reported and outdated. We therefore conducted a 12-month prospective survey to assess the aetiology and antifungal resistance patterns among symptomatic Ecuadorian patients. VVC diagnosis was confirmed by microscopic examination and culture. Isolates were identified by biochemical and molecular methods. In vitro antifungal susceptibilities to amphotericin B, clotrimazole, fluconazole, itraconazole, miconazole, and nystatin were determined by CLSI methods. Among 195 women, 71 VVC episodes were recorded (36.4%), whereof 56 (28.7%) had acute VVC (AVVC) and 15 (7.7%) had recurrent VVC (RVVC). The predominant species was *Candida albicans*, isolated in pure culture from 45 AVVC (80.3%) and 9 RVVC patients (60%), and in mixed culture from 7 AVVC (12.5%) and 3 RVVC patients (20%). *Candida glabrata* and *Saccharomyces cerevisiae* were also isolated in AVVC and RVVC patients, but *Candida parapsilosis* and *Candida famata* were only isolated from AVVC. Fluconazole- and miconazole-resistant *C. albicans* isolates were recovered from 5 (8.9%) and 24 (42.9%) of 56 AVVC patients, respectively, and from 1 (8.3%) and 5 (41.7%) of 12 RVVC patients, respectively. Fluconazole and miconazole resistance is relevant in Ecuador, emphasising the need for targeted antifungal strategies.

## 1. Introduction

Vulvovaginal candidiasis (VVC) is a prevalent fungal infection, primarily caused by species of *Candida*. It affects up to 75% of women of reproductive age at least once in their lifetime. Acute VVC (AVVC) is the second-most common vaginal infection globally, with peak incidence among women aged 15–45 years and during the third trimester of pregnancy. Symptoms such as genital itching, pain, increased vaginal discharge, fatigue, irritability, insomnia, memory impairment, headaches, and depression can significantly disrupt daily life and intimate relationships. AVVC is a multifactorial condition, more frequent in women with immunosuppression, pregnancy, diabetes mellitus, prolonged antibiotic use, hormone therapy (including oestrogens and hormone replacement), bacterial or viral coinfections, and extended use of broad-spectrum antimicrobials. These factors often lead to genital dysbiosis, particularly affecting lactobacilli populations and promoting *Candida* overgrowth [1,2,3,4].

Approximately 9% of affected women develop recurrent VVC (RVVC), which imposes a considerable physical, psychological, and social burden. RVVC is a major clinical challenge due to its global prevalence and associated healthcare costs, underscoring the need for effective therapeutic and preventive strategies. An estimated 138 million women worldwide suffer from RVVC, with the highest burden among those aged 25–34 years [5]. While *Candida albicans* remains the predominant pathogen in both AVVC and RVVC, non-*C. albicans* species are increasingly involved [2,3,6].

Despite its frequency, diagnosis is often empirical and symptom-based. Microscopy is recommended for AVVC, whereas culture with species identification and antifungal susceptibility testing is essential for RVVC. A key challenge in VVC management is the delayed or inaccurate identification of *Candida* species. Phenotypic methods may fail to detect non-*C. albicans* species, and in vitro antifungal susceptibility testing can take several days, delaying appropriate treatment and increasing morbidity and costs. Misdiagnosis is common due to overlapping symptom with infections caused by *Trichomonas vaginalis*, *Gardnerella vaginalis*, and other vaginal pathogens [3,7]. Accurate laboratory diagnosis is therefore crucial to guiding effective therapy and preventing antifungal resistance [4,8,9].

Standard treatment includes polyenes and azoles. Azoles—particularly clotrimazole, miconazole, and fluconazole—are widely used due to their affordability and low toxicity. However, inappropriate or prolonged use has led to the emergence of resistant *Candida* isolates [3,4,10,11].

Although previous studies have reported VVC prevalence rates exceeding 20%, data on the epidemiology of VVC in Ecuador remain limited and outdated [12,13,14,15,16,17]. To address this gap, we conducted a 12-month prospective study to assess the aetiology and antifungal resistance patterns among symptomatic Ecuadorian women. This study evaluates the species distribution and in vitro susceptibility profiles of *Candida* isolates, focusing on commonly used antifungal drugs: two polyenes (amphotericin B and nystatin) and four azoles (clotrimazole, fluconazole, itraconazole, and miconazole).

## 2. Patients, Materials, and Methods

### 2.1. Study Design

A prospective cross-sectional study was conducted to determine the prevalence of *Candida* and other medically relevant fungi as aetiological agents of vulvovaginal infections in women presenting with genital symptoms suggestive of VVC at the Medical Service of the Pontifical Catholic University of Ecuador (PUCE) in Quito. Women who voluntarily participated were attended by gynaecologists, who explained the study objectives and obtained informed consent. The study was approved by the PUCE Ethics Committee (CEISH-375-2017).

Inclusion criteria were women aged 18 years or older, presenting with symptoms consistent with VVC, and having signed the informed consent form. Exclusion criteria included: being under 18, pregnant, breastfeeding, menstruating, or having received antifungal treatment within the previous month. Following consent, a clinical protocol was implemented, including a full gynaecological examination performed at least three days after the end of menstruation, to allow clinical classification. Gynaecological specimens were collected using swabs with appropriate transport medium from the outer third and fornix of the vagina, as well as the vulva, prior to any diagnostic procedures.

AVVC was defined as an episode of vulvovaginal inflammation with pruritus, pain, and/or leucorrhoea, accompanied by microscopic evidence of blastoconidia, pseudohyphae, or hyphae consistent with *Candida*, and/or a positive culture. Episodes caused by non-*C. albicans* species and/or occurring in pregnant women, immunocompromised individuals, or those with poorly controlled diabetes were classified as complicated VVC. Only cases of complicated VVC caused by non-*C. albicans* species were included in the study; other forms—such as those occurring in pregnant women, immunocompromised individuals, or patients with poorly controlled diabetes—were excluded.

RVVC was defined as four or more episodes within one year, with at least two episodes confirmed by microscopy and at least one by pure culture isolation of *Candida* or other fungi.

Based on these criteria, vaginal specimens were collected from 195 women aged 18 to 67 years (mean age: 26.8 years) over a 12-month period (1 October 2021 to 30 September 2022).

### 2.2. Microscopic Examination, Culture, Isolation, and Identification of Candida and Other Fungi from Patients with Vulvovaginitis

Fresh microscopic examination of all vaginal specimens was performed using saline solution or 10% potassium hydroxide to detect *Candida* blastoconidia, pseudohyphae, or hyphae. Additionally, Gram-stained samples were examined to visualise *Candida* cells, which appear dark violet or deep blue under light microscopy.

Clinical specimens were cultured on CHROMagar Candida (CHROMagar, Paris, France), a chromogenic medium designed for the isolation and presumptive identification of *Candida* species based on enzymatic hydrolysis of chromogenic substrates such as β-galactosidase and hexosaminidases. The colony chromatic characteristics of the most prevalent species were green for *C. albicans*, creamy white or mauve for *Candida glabrata* (*Nakaseomyces glabratus*), blue for *Candida tropicalis*, and pink with a whitish edge for *Candida krusei* (*Pichia kudriavzevii*) [18,19].

All laboratory procedures were performed using pure yeast isolates obtained from single colonies grown on chromogenic agar. These isolates were subsequently cultured on Sabouraud dextrose agar for phenotypic and molecular identification, as well as for antifungal susceptibility testing. Species identification was carried out using conventional mycological techniques, including germ tube production in serum, chlamydoconidia formation, and biochemical profiling. Chlamydoconidia formation and microscopic morphology were assessed on cornmeal agar supplemented with Tween 80 using the Dalmau plate technique. Biochemical identification was performed using the API ID 32C system, which evaluates carbon source assimilation across a panel of dehydrated substrates [19,20]. Species identification was confirmed by PCR using species-specific primers to species belonging to *C. albicans* complex (CR-f and Cr-r), targeting the *HWP1* gene [21]. Species belonging to the *C. glabrata* complex were identified by multiplex-polymerase chain reaction (multiplex-PCR) using primers GLA-f, NIV-f, BRA-f, and UNI-5.8S, targeting the internal transcribed spacer (ITS1) region and 5.8S rRNA gene [22]. For species belonging to *C. parapsilosis* complex, identification was performed using polymerase chain reaction-restriction fragment length polymorphism (PCR-RFLP) with S1-F forward and S1-F reverse primers targeting the secondary alcohol dehydrogenase (*SADH*) gene region [23].

Reference strains used for molecular identification and antifungal susceptibility testing were obtained from the American Type Culture Collection (ATCC, USA), the National Collection of Yeast Cultures (NCYC, Norwich, UK), and the Centraalbureau voor Schimmelcultures (CBS, Utrecht, The Netherlands). These included *C. albicans* ATCC 64548 and ATCC 64550, *Candida bracarensis* (*Nakaseomyces bracarensis*) NCYC 3133, *C. glabrata* ATCC 90030, *C. krusei* ATCC 6258, *Candida metapsilosis* ATCC 96144, *Candida nivariensis* (*Nakaseomyces nivariensis*) CBS 9984, *Candida orthopsilosis* ATCC 96141, and *Candida parapsilosis sensu stricto* ATCC 22019.

### 2.3. Genotyping of Candida albicans Isolates

To investigate the molecular epidemiology and genotypic relationships among vaginal *C. albicans* isolates, genotyping was performed on 70 isolates: 62 from patients with AVVC and 8 from those with RVVC.

Genomic DNA was extracted by directly resuspending a colony from a 24–48 h pure culture into a 20 µL PCR reaction mixture containing the following: 8.2 µL of Biomix DNA Polymerase master mix (Bioline, London, UK), which includes reaction buffer, dNTPs, magnesium chloride, and Taq DNA polymerase; 0.15 µL of forward primer (CA-INT-L: ATA AGG GAA GTC GGC AAA ATA GAT CCG TAA); 0.15 µL of reverse primer (CA-INT-R: CCT TGG CTG TGG TTT CGC TAG ATA GTA GAT); and 11.5 µL of sterile MilliQ double-distilled water.

PCR amplification was carried out using a GeneAmp PCR System 9700 thermal cycler (Applied Biosystems, Foster City, CA, USA) under the following conditions: initial denaturation at 96 °C for 120 s; 30 cycles of denaturation at 96 °C for 30 s, annealing at 65 °C for 5 s, and extension at 74 °C for 30 s; followed by a final extension at 75 °C for 15 min.

PCR products were resolved by electrophoresis for 70 min at 90 V in a horizontal Sub-cell GT chamber (Bio-Rad Laboratories, Hercules, CA, USA) using 1.5% low electroendosmosis (EEO) agarose gel stained with ethidium bromide. A Hyperladder IV molecular weight marker (Bioline), comprising ten bands ranging from 100 to 1000 base pairs, was used to estimate fragment sizes. Gel visualisation was performed using the Gel Chemidoc imaging system (Bio-Rad). Based on PCR fragment sizes, *C. albicans* genotypes were classified as follows: genotype A, 450 bp; genotype B, 840 bp; genotype C, 450 and 840 bp; genotype D, 1080 bp; and genotype E, 1400 bp [24].

### 2.4. In Vitro Antifungal Susceptibility Testing

Antifungal susceptibility of *Candida* isolates was assessed using two standardised methods: the agar disc diffusion method and the broth microdilution method, following Clinical and Laboratory Standards Institute (CLSI) guidelines. These complementary approaches allowed for the evaluation of susceptibility profiles against commonly used antifungal agents, ensuring both qualitative and quantitative assessment of resistance patterns.

#### 2.4.1. Disc Diffusion Susceptibility Testing According to CLSI M44 Guidelines

A total of 70 *Candida* isolates were evaluated using a modified version of the CLSI M44-A2 disc diffusion method [25]. These included 70 *C. albicans* isolates, 7 *C. glabrata* isolates, and 1 *C. parapsilosis* isolate. Antifungal susceptibility testing was performed using Neo-Sensitabs antifungal tablets (Rosco Diagnostica, Albertslund, Denmark), containing the following concentrations: 50 μg of nystatin, 10 μg of clotrimazole, and 10 μg of miconazole.

Fungal suspensions were prepared in sterile saline and adjusted to a 0.5 McFarland standard (approximately 1–5 × 10^6^ CFU/mL) using colonies grown on Sabouraud dextrose agar at 37 °C for 24 h. Each inoculum was evenly spread onto Mueller–Hinton agar (Difco; Becton, Dickinson and Co., Franklin Lakes, NJ, USA) supplemented with 2% (*w*/*v*) glucose and 0.5 μg/mL methylene blue using sterile cotton swabs. After allowing the surface to dry for five minutes, antifungal tablets were placed on the agar, and plates were incubated at 37 °C for 24 h.

Following incubation, inhibition zone diameters were measured in millimetres using a calliper, excluding microcolonies at the halo edge or large colonies within the inhibition zone. If no growth was observed, incubation was extended for an additional 24 h. Isolates were classified according to their in vitro susceptibility to each antifungal agent.

Quality control was performed using *C. krusei* ATCC 6258 and *C. parapsilosis* ATCC 22019 reference strains.

#### 2.4.2. In Vitro Antifungal Susceptibility Testing Using the CLSI M27 Broth Microdilution Method in RPMI

In addition to the disc diffusion assay, the in vitro susceptibility of 78 vaginal isolates to amphotericin B, fluconazole, and itraconazole (Sigma-Aldrich, St. Louis, MO, USA) was evaluated using the CLSI reference broth microdilution method (M27-A3 and M27-S4) [26,27]. Amphotericin B was included as the reference comparator (gold standard) for the in vitro antifungal activity of the other drugs evaluated in this study.

Microtitre plates were prepared from a fluconazole stock solution (6400 μg/mL) diluted in sterile water via two-fold serial dilutions. Each dilution was added to RPMI-1640 medium buffered to pH 7.0 with 0.165 M morpholinepropanesulphonic acid (MOPS), yielding final concentrations of 0.25–128 μg/mL. Aliquots of 100 μL were dispensed into 96-well U-bottom microplates, with increasing concentrations from columns 1 to 10. Columns 11 and 12 contained 100 μL of drug-free RPMI medium as growth and sterility controls, respectively. Plates were labelled and stored at −70 °C for up to six months. The inoculum was prepared by diluting a yeast suspension equivalent to a 0.5 McFarland standard in 0.85% saline to achieve a final concentration of 1–5 × 10^3^ CFU/mL in RPMI 1640. Each well in the drug-containing columns and the growth control column received 100 μL of the inoculum, resulting in final drug concentrations of 0.125–64 μg/mL and an inoculum density of 0.5–2.5 × 10^3^ CFU/mL. Plates were incubated at 37 °C and examined at 24 and 48 h. Quality control was performed using *C. krusei* ATCC 6258 and *C. parapsilosis* ATCC 22019. Growth in each well was visually compared to the growth control. Microtitre plates for amphotericin B and itraconazole were prepared from 3200 μg/mL stock solutions, following the same protocol as for fluconazole.

The minimum inhibitory concentration (MIC) was defined as the lowest antifungal concentration resulting in ≥50% growth inhibition at 24 h compared to the drug-free control. Isolates were classified according to Clinical Breakpoints (CBPs) and Epidemiological Cutoff Values (ECVs), which are sensitive indicators for detecting emerging resistance. Isolates were categorised as wild-type (WT; no acquired resistance) or non-wild-type (NWT; with resistance mechanisms). CLSI interpretive breakpoints were applied for fluconazole and itraconazole. As no CLSI breakpoints exist for amphotericin B, isolates inhibited at ≤1 μg/mL were considered susceptible, and those at ≥2 μg/mL were deemed resistant.

### 2.5. Statistical Analysis

Most of the data were descriptive, and standard variables commonly used in culture, identification, and in vitro antifungal susceptibility testing were employed. MICs, MIC ranges, geometric mean (GM) MICs, MIC_50_, and MIC_90_ values—representing the concentrations that inhibited 50% and 90% of isolates, respectively—were also calculated. Data analysis was performed using GraphPad Prism (version 5.0, GraphPad Software, La Jolla, CA, USA) and SPSS (version 21.0, IBM, Armonk, NY, USA). Depending on data distribution, either parametric or non-parametric tests were applied. Student’s *t*-test was used for normally distributed data. For non-normally distributed data, the Kruskal–Wallis test with Dunn’s post hoc test and the Mann–Whitney *U* test were employed. Analysis of variance (ANOVA) was used for comparisons between groups. In all analyses, *p* < 0.05 was considered statistically significant.

## 3. Results

### 3.1. Aetiology of Vulvovaginal Candidiasis

Approximately 71 of the 195 patients included in the study were diagnosed with VVC. Table 1 presents detailed sociodemographic data for all participants. Of these, 56 patients (28.7%) experienced an AVVC, while 15 (7.7%) had a new episode consistent with a pattern of RVVC. Nine of the fifty-one patients with AVVC had mixed infections with *G. vaginalis* (17.6%), and one patient had a mixed infection with *T. vaginalis* (1.9%). No fungal elements were detected microscopically or isolated in cultures from the vaginal specimens of the remaining 124 women (63.6%). Their symptoms were attributed to other vaginal conditions, including bacterial vaginosis (27 patients, 21.8%), trichomoniasis (37 patients, 29.8%), both conditions concurrently (6 patients, 4.8%) or to non-infectious genital conditions (54 patients, 43.6%). The mean age of patients presenting with vulvovaginal symptoms but without candidiasis was 24.5 years (range: 18–58 years). A comparable mean age of 26.8 years was observed among those diagnosed with VVC (range: 18–67 years). Within this group, patients with AVVC had a mean age of 25.8 years (range: 18–67 years), while those with RVVC exhibited a slightly higher mean age of 30.5 years (range: 21–45 years). The mean age of patients presenting with vulvovaginal symptoms but without candidiasis was 24.5 years (range: 18–58 years).

Among the study participants, 11 of the 124 women without VVC (9.9%) were receiving antibiotic treatment. Similar proportions were observed in the AVVC group (4 of 56; 7.1%) and the RVVC group (1 of 15; 6.7%). Hormonal contraceptive use was reported by 82 women without VVC (66.1%), 27 with AVVC (48.2%), and 7 with RVVC (46.7%). Of the 71 women with VVC, 26 (36.6%) reported having a single sexual partner, while 45 (63.4%) had multiple partners. Among those with AVVC, 38 (67.9%) reported two or more sexual partners—a significantly higher proportion (*p* < 0.0001) than among women with RVVC (7; 46.7%) or those without VVC (26; 20.9%). Pure cultures were obtained from clinical samples of 49 patients with AVVC (87.5%), and mixed cultures from seven (12.5%). Table 2 summarises the culture results from the 56 patients with acute vulvovaginal candidiasis.

The predominant species was *C. albicans*, isolated in pure culture from 45 patients (80.3%) and in mixed culture from 7 (12.5%). *C. glabrata* was isolated in pure culture from two patients (3.6%), while *C. parapsilosis* and *Saccharomyces cerevisiae* were each isolated in pure culture from one patient (1.8%). *C. glabrata* was also detected in two mixed cultures (3.6%), and *S. cerevisiae* in three (5.3%), all in combination with *C. albicans*.

Additionally, *Candida famata* (*Debaryomyces hansenii*) was isolated in two mixed cultures with *C. albicans* (3.6%).

Among the 15 patients with RVVC, 12 (80%) yielded pure cultures and three (20%) mixed cultures. Fungal diversity was lower in this group, with only *C. albicans*, *C. glabrata*, and *S. cerevisiae* being isolated.

*C. parapsilosis* and *C. famata* were not detected. *C. albicans* remained the predominant species, isolated in pure culture from nine patients (60%) and in mixed culture from three (20%). *S. cerevisiae* was isolated in pure culture from two patients (13.3%) and *C. glabrata* from one (6.7%). Additionally, *S. cerevisiae* was found in two mixed cultures (13.3%) and *C. glabrata* in one (6.7%), both in combination with *C. albicans*.

A total of 70 *C. albicans* isolates were genotyped. Genotype A was the most prevalent, identified in 65 isolates (92.9%) across both patient groups. It was the sole genotype detected among patients with RVVC. Among those with AVVC, four isolates (6.5%) belonged to genotype B, and one isolate (1.6%) to genotype C.

### 3.2. In Vitro Antifungal Susceptibility Testing

Antifungal susceptibility testing revealed fungal isolates with reduced susceptibility to azole antifungal agents, particularly fluconazole and miconazole (Table 3 and Table 4).

Nystatin demonstrated activity against all tested isolates. Sixty-eight of the seventy *C. albicans* isolates (97.1%) were susceptible to amphotericin B. One isolate from a patient with AVVC (1.7%) and another from a patient with RVVC (8.3%) were resistant. However, no statistically significant differences in resistance profiles were observed (*p* > 0.05). Table 4 presents the MIC distribution of amphotericin B against *C. albicans* isolates. Among AVVC isolates, the MIC_50_ was 0.125 µg/mL, the MIC_90_ was 0.25 µg/mL, and the MIC range was 0.03–4 µg/mL, with a GM MIC of 0.135 µg/mL. Similar results were observed for RVVC isolates MIC_50_ of 0.125 µg/mL, MIC_90_ of 0.125 µg/mL, MIC range of 0.03–0.25 µg/mL, and GM MIC of 0.114 µg/mL. Two *C. albicans* genotype A isolates were resistant to amphotericin B. None of the four genotype B isolates nor the single genotype C isolate from AVVC patients exhibited resistance. For genotype A isolates from AVVC patients, the MIC_50_ was 0.125 µg/mL, MIC_90_ was 0.25 µg/mL, MIC range was 0.03–4 µg/mL, and GM MIC was 0.125 µg/mL. Genotype A isolates from RVVC patients showed similar results: MIC_50_ of 0.125 µg/mL, MIC_90_ of 1 µg/mL, MIC range of 0.03–4 µg/mL, and GM MIC of 0.194 µg/mL. For genotype B isolates, the MIC_50_ and MIC range were both 0.125 µg/mL, with a GM MIC of 0.21 µg/mL. The single genotype C isolate had an MIC of 0.125 µg/mL.

Among AVVC patients, *C. glabrata* isolates were inhibited at an amphotericin B MIC_50_ of 0.25 µg/mL and a GM MIC of 0.297 µg/mL, with a range of 0.25–0.5 µg/mL. In RVVC patients, *C. glabrata* isolates were inhibited at an MIC_50_ of 0.06 µg/mL and a GM MIC of 0.243 µg/mL, with a range of 0.06–4 µg/mL.

Sixty-two *C. albicans* isolates (88.6%) were susceptible to clotrimazole, as were all *C. glabrata* isolates. Eight *C. albicans* isolates (11.4%) showed reduced susceptibility, and the *C. parapsilosis* isolate exhibited intermediate susceptibility. The proportion of *C. albicans* isolates with reduced susceptibility to clotrimazole was significantly higher among RVVC patients (33.3%) compared to AVVC patients (6.9%). One patient from each group was infected with a clotrimazole-resistant *C. albicans* isolate.

Thirty-three of the seventy *C. albicans* isolates (47.1%) showed reduced susceptibility to miconazole, with twenty-nine isolates (41.4%) classified as resistant. All *C. glabrata* isolates and 37 *C. albicans* isolates were susceptible, while the *C. parapsilosis* isolate showed intermediate susceptibility. The proportion of *C. albicans* isolates with reduced susceptibility to miconazole was not significantly higher in RVVC patients (58.4%) compared to AVVC patients (44.8%). Among these, 24 AVVC patients and 5 RVVC patients were infected with miconazole-resistant *C. albicans* isolates.

Of the 70 *C. albicans* isolates, 53 (75.7%) were susceptible to fluconazole in vitro (Table 3). Seventeen isolates (24.3%) exhibited reduced susceptibility; five from patients with AVVC (8.6%) and one from a patient with RVVC (8.3%) were classified as resistant. No statistically significant differences were observed between isolates from acute and recurrent episodes (*p* > 0.05). Additionally, nine isolates from AVVC cases (15.5%) and two from RVVC cases (16.7%) were classified as dose-dependent susceptible. Again, no significant differences were found between the two groups (*p* > 0.05). All seven *C. glabrata* isolates and the single *C. parapsilosis* isolate were susceptible to fluconazole.

Table 4 presents the MIC distribution of fluconazole against *C. albicans* isolates. For AVVC cases, the MIC_50_ was 0.125 µg/mL, the MIC_90_ was 4 µg/mL, and the MIC range was 0.125–16 µg/mL, with a GM MIC of 0.444 µg/mL. Similar values were observed for RVVC cases: MIC_50_ of 0.250 µg/mL, MIC_90_ of 4 µg/mL, MIC range of 0.125–8 µg/mL, and GM MIC of 0.530 µg/mL.

Among genotype A *C. albicans* isolates, 48 were susceptible to fluconazole. However, five isolates from AVVC cases (8.8%) and one from an RVVC case (12.5%) were resistant. All three genotype B isolates and the single genotype C isolate from AVVC cases were susceptible, except for one genotype B isolate, which was resistant. For AVVC cases, genotype A isolates showed an MIC_50_ of 0.125 µg/mL, MIC_90_ of 4 µg/mL, MIC range of 0.125–16 µg/mL, and GM MIC of 0.444 µg/mL. For RVVC cases, genotype A isolates had an MIC_50_ of 0.125 µg/mL, MIC_90_ of 4 µg/mL, MIC range of 0.125–8 µg/mL, and GM MIC of 0.530 µg/mL. The GM MIC for genotype B isolates was 2.18 µg/mL, with an MIC_50_ of 4.16 µg/mL and a range of 0.125–8 µg/mL. The MIC for the genotype C isolate was 0.125 µg/mL.

Among *C. glabrata* isolates from AVVC cases, the MIC_50_ and GM MIC for fluconazole were 2 µg/mL and 1.414 µg/mL, respectively, with a range of 0.5–2 µg/mL. For RVVC cases, the MIC_50_ was 0.125 µg/mL and the GM MIC was 0.157 µg/mL, with a range of 0.125–0.25 µg/mL. No significant differences were found between the 65 *C. albicans* isolates from acute and recurrent cases (*p* > 0.05).

*Candida albicans* exhibited 100% susceptibility to itraconazole in vitro (Table 3 and Table 4). For AVVC cases, the MIC_50_ was 0.03 µg/mL, the MIC_90_ was 0.25 µg/mL, and the MIC range was 0.03–0.5 µg/mL, with a GM MIC of 0.055 µg/mL. Similar results were observed for RVVC cases: MIC_50_ of 0.03 µg/mL, MIC_90_ of 0.125 µg/mL, MIC range of 0.03–0.25 µg/mL, and GM MIC of 0.045 µg/mL. All *C. albicans* isolates belonging to genotypes A, B, and C were susceptible to itraconazole in vitro. For genotype A isolates from AVVC patients, the MIC_50_ was 0.03 µg/mL, MIC_90_ was 0.25 µg/mL, MIC range was 0.03–0.5 µg/mL, and GM MIC was 0.057 µg/mL. Genotype A isolates from RVVC patients showed similar results: MIC_50_ of 0.03 µg/mL, MIC_90_ of 0.25 µg/mL, MIC range of 0.03–0.25 µg/mL, and GM MIC of 0.041 µg/mL. For the four genotype B isolates, the MIC_50_ was 0.03 µg/mL, the MIC range was 0.03–0.06 µg/mL, and the GM MIC was 0.03 µg/mL. The single genotype C isolate had an MIC of 0.03 µg/mL.

All *C. glabrata* and *C. parapsilosis* isolates were also susceptible to itraconazole. Among *C. glabrata* isolates from AVVC cases, the MIC_50_ was 0.030 µg/mL, the MIC range was 0.030–0.125 µg/mL, and the GM MIC was 0.043 µg/mL. For RVVC cases, both the MIC_50_ and GM MIC were 0.030 µg/mL.

## 4. Discussion

AVVC is one of the most common reasons for medical consultation worldwide [25]. In Ecuador, it is a frequent cause of vaginal infections, accounting for approximately 20–30% of such cases. However, studies that accurately determine the incidence of this mycosis in patients presenting with genitourinary symptoms are limited [12,13,14,15,16,17]. In many instances, as in other countries, antifungal treatment is initiated based solely on clinical diagnosis, without microbiological confirmation. One reason for the lack of precise data is that VVC is not a notifiable disease. Moreover, many diagnoses are based on symptoms alone, whether by healthcare professionals or the patients themselves, often followed by empirical treatment or self-medication. However, the symptoms are non-specific and may also be observed in other genitourinary conditions such as bacterial vaginosis or trichomoniasis. This situation often leads to pharmacological treatment that may be unnecessary in many cases [4].

The global incidence of VVC among women with genital symptoms ranges from 12% to 75%. The highest incidences of VVC have been reported in studies conducted in Africa and Asia, exceeding 30% [28,29,30,31,32,33,34,35,36], while the lowest rates are observed in European countries and Canada (10–20%) [37,38,39,40]. *C. albicans* remains the most frequently isolated species, but in recent decades, a shift in aetiology has been observed, with increasing incidence of non-*C. albicans* species [41,42,43]. In the current study, 71 of the 195 participants (36.4%) were diagnosed with VVC based on confirmed clinical and microbiological criteria: 56 (28.7%) had an acute episode and 15 (7.7%) experienced a new episode within a recurrent pattern. This incidence lies between those reported in Europe and Africa and it is comparable to rates reported in other American countries [2,44,45,46,47]. In Ecuador, studies on the prevalence of candidiasis, particularly VVC, report a wide range, from 7% to 80%, depending on the region [12,13,14,15,16,17]. These findings must be interpreted with caution, as most studies lack access to definitive microbiological diagnostic methods.

Vaca et al. [12], in a study of 213 adolescent girls in a tropical region of Ecuador, found that 23.7% had VVC caused by *C. albicans*. However, this study did not specify whether the participants were symptomatic, nor did it include fungal culture. In coastal Ecuador, Ávila Tandazo et al. [15] reported a prevalence of VVC of 45% among 506 pregnant women. The most frequently isolated species were *C. albicans* (72%) and *C. glabrata* (19%). In the inner province of Azuay, Orellana et al. [16] reported the isolation of 136 *Candida*: *C. albicans* (92.6%), *C. glabrata* (6.6%), and *C. parapsilosis* (0.7%). However, the study did not specify the diagnostic methods used for species identification, nor did it report the prevalence rate of VVC. The incidence of RVVC reported in Ecuadorian studies is 7.7%, which is similar to that described in other regions of the world, ranging from 5% to 10%. Zurita et al. [13] estimated that 308,000 women in Ecuador suffer from RVVC each year.

*C. albicans* remains the most frequently isolated. Species such as *C. glabrata*, *C. tropicalis*, *C. parapsilosis*, and *C. krusei* are isolated far less frequently. In most cases of VVC, a single species is isolated from clinical cultures. However, between 1% and 10% of cultures are mixed, with two or more species present [38,40]. In the current study, 49 pure monomicrobial cultures (77.8%) and 14 mixed cultures (22.2%) were obtained from vaginal samples of patients with AVVC. These findings highlight the importance of accurately identifying the species of *Candida* involved in mixed infections to ensure appropriate antifungal therapy. The predominant species was *C. albicans*, which was isolated in pure culture from 45 patients (80.3%) and in mixed culture from seven patients (12.5%). Additionally, in two women with AVVC, *C. glabrata* was isolated (3.6%), and *C. parapsilosis* and *S. cerevisiae* were each isolated in pure culture from one patient (1.8%). In mixed cultures, *C. glabrata* was found in two cases (3.6%) and *S. cerevisiae* in three cases (5.3%), all in combination with *C. albicans*. *C. famata* was also isolated in two mixed cultures with *C. albicans* (3.6%).

Among the 15 patients with RVVC, 12 (80.0%) yielded pure cultures and 3 had mixed cultures. Fungal species diversity was lower in this group, with only *C. albicans*, *C. glabrata*, and *S. cerevisiae* being isolated: *C. parapsilosis* and *C. famata* were not detected. *C. albicans* was again the predominant species, isolated in pure culture from nine patients (60.0%) and in mixed culture from three (20.0%), in combination with *S. cerevisiae* in two patients (13.3%) and *C. glabrata* in one patient (6.7%).

Molecular confirmation of vaginal *Candida* isolates was essential to exclude cryptic species within the *Candida albicans*, *Candida glabrata*, and *Candida parapsilosis* complexes, which may exhibit distinct pathogenic traits and in vitro antifungal susceptibility profiles [21,22,23]. The isolation of non-*C. albicans Candida* species has been associated with the widespread and potentially inappropriate use of azole antifungal drugs, particularly self-medication with over-the-counter (OTC) treatments for presumed AVVC. This antifungal selective pressure may favour species such as *C. glabrata*, which exhibit lower susceptibility to azoles, or *C. krusei*, intrinsically resistant to fluconazole [34,36,38,43]. This is further facilitated by the availability of OTC topical antifungal drugs without prescription in many countries [3,4]. The development of rapid diagnostic methods, potentially usable at home, could reduce misdiagnosis, avoid unnecessary treatments, and enable more appropriate antifungal therapy.

In the present study, non-*C. albicans* species accounted for 6.3% of pure cultures in AVVC and 16.7% in RVVC. Their presence in mixed cultures was also lower in both groups (11.1% in acute vs. 16.7% in recurrent cases). *C. glabrata* and other non-*C. albicans* species has been associated with advancing patient age. Most studies report the combination of *C. albicans* and *C. glabrata* as the most frequent mixed infection [38,48], but we found no previous reports describing the association between *C. albicans* and *S. cerevisiae*. The aetiological role of non-*C. albicans* species and *S. cerevisiae* in VVC is questioned. However, these species are increasingly isolated from women with acute infections, and in mixed infections with *C. albicans*, a synergistic pathogenic effect may occur [48,49,50,51,52].

*C. glabrata* is the most common non-*C. albicans* species in VVC in Ecuador, with prevalence ranging from 6% to 20% [15,16,53]. While single-species infections are more common, 1–10% of women may present with mixed infections, most frequently involving *C. albicans* and *C. glabrata* [36,38]. There is growing evidence of increased colonisation, invasion of vaginal epithelium, and significant infection by *C. glabrata* in co-infection with *C. albicans* [50].

*S. cerevisiae* has been reported in 1–6% of cases and rarely causes vaginitis. Some studies suggest that its virulence may be due to its association with *C. albicans* as a co-pathogen, similar to *C. glabrata* [49,54]. *C. famata* is an uncommon cause of vulvovaginal candidiasis, with reported isolation rates ranging from 0.2% to 2% in surveillance studies, as in the study conducted in the Toronto–Hamilton area, Ontario, Canada [55]—figures that are consistent with those found in this study.

*C. parapsilosis* showed a prevalence of 1.4%, which aligns with previous studies conducted in Ecuador reporting rates between 0.7% and 1% [16]. Similarly low percentages have been reported in other countries among patients with AVVC [37,39,40,56].

In the current study, 70 *C. albicans* isolates were genotyped. Sixty-five isolates (92.9%) belonged to genotype A, which is the predominant genotype reported in all published studies on VVC [57,58,59,60,61]. Genotypes B (5.7%) and C (1.4%) were less frequent and were only isolated from patients with AVVC. Díaz-Huerta et al. [61] have found a similar distribution of *C. albicans* genotypes (A: 82.6%; B: 17.4%) in vaginal isolates of Mexican women suffering from vulvovaginitis. Genotype A isolates have been shown to be more virulent in vitro than genotypes B and C, but the results are controversial [58,61]. Fan et al. [55] observed that *C. albicans* genotypes were associated with the severity of VVC, although this relationship was not confirmed by Güzel et al. [59]. This contrast may be attributable to differing clinical characteristics of the patients included in each study, to geographical variations among the *C. albicans* isolates analysed, or both.

VVC is associated with multiple, not fully understood, risk factors. It is more frequent in individuals with immune disorders, women undergoing prolonged antibiotic or hormone therapy, and those who are sexually active. These factors may lead to endogenous changes in the genital microbiota, particularly in the composition of lactobacilli. VVC is most common in women aged 20 to 40 years, which is considered the age group with the highest reproductive potential [3,4,38]. In the present study, the mean age of patients with vulvovaginal symptoms but without candidiasis was 24.5 years, while it was 26.5 years among those with VVC. Women with acute episodes were younger (25.8 years) than those with recurrent infections (30.5 years).

Antibiotic treatment, particularly with broad-spectrum antibiotics, can disrupt the vaginal microbiota and promote both fungal colonisation and the development of vulvovaginal candidiasis [3,4,42,48,62]. In our study, this factor could not be associated with the development of candidiasis, as the proportion of women who had received antibiotics in the previous month was low: 9.9% among those without candidiasis, 7.1% among those with AVVC, and 6.7% among those with RVVC.

The use of hormonal contraceptives has been associated with the development of VVC [38,48]. However, there is considerable debate and controversy regarding the actual impact of these agents on the disease. The lack of consensus may be due to differences in the composition and hormone concentrations of contraceptive formulations, as well as the limited detail in epidemiological data. In our study, no association was found between oral contraceptive use and VVC. In fact, hormonal contraceptive use was more common among women with genitourinary symptoms who did not have VVC: 82 women without VVC (66.1%) used hormonal contraceptives, compared to 27 women with AVVC (48.2%) and 7 with recurrent candidiasis (46.7%). However, detailed information on the specific contraceptive formulations used was not available, but it is likely that most were low-dose hormonal contraceptives.

Although VVC is not classified as a sexually transmitted infection, certain sexual behaviours have been associated with an increased frequency of episodes. Frequent sexual intercourse may increase vaginal *Candida* concentrations through microtrauma to the vaginal mucosa, modulation of local immune responses, pH alkalinisation, and disruption of the microbiota due to semen, which contains high levels of antigens, antibodies, and cytokines. Oral sex may also influence vaginal microbiota and pH by introducing saliva, creating conditions favourable for fungal overgrowth and subsequent infection. In the current study, a statistically significant association was observed between a higher number of sexual partners and the presence of VVC. Among women with genital symptoms but without VVC, 20.9% reported having two or more sexual partners. This proportion was much higher among women with AVVC (67.9%) and those with RVVC (46.7%). Several studies have reported that young women with multiple sexual partners are at increased risk of VVC, although this association is not consistently observed across all studies [35,37].

The therapeutic protocol followed at Medical Service of the PUCE, based on international guideline recommendations [4,20], was as follows: For patients with AVVC, oral treatment with 150 mg fluconazole was administered as an initial dose, followed by a second dose after 72 h. This was accompanied by topical treatment with vaginal ovules containing either 200 mg clotrimazole or 400 mg miconazole for seven days. As an alternative to these topical azoles, a daily vaginal tablet of nystatin for seven days was used in complicated AVVC. For patients with RVVC, a suppression regimen was implemented with oral fluconazole 150 mg on days 1, 4, and 7, followed by weekly maintenance therapy for six months. During this maintenance period, patients also received a weekly vaginal ovule containing either 500 mg clotrimazole or 400 mg miconazole. As an alternative to these topical azoles, a weekly vaginal tablet of nystatin was also used. Oral itraconazole was occasionally used in the treatment of VVC that were refractory to oral fluconazole.

Nystatin has demonstrated strong activity against various *Candida* species [63]. In the current study, this antifungal agent was also highly effective against all *Candida* isolates tested, consistent with findings from a study conducted in Ecuador by Ávila Tandazo et al. [15]. Nystatin remains one of the preferred therapeutic options in Ecuador for treating complicated AVVC and RVVC, particularly in cases where *C. albicans* and *C. glabrata* isolates have shown resistance to fluconazole. Amphotericin B has also a broad antifungal spectrum. In the present study, 97.1% of isolates were susceptible, consistent with most previous studies reporting low resistance rates [34,38,40]. Resistance to amphotericin B is rare but may occur in isolates with cell wall modifications or a high capacity for biofilm formation, which produces a dense extracellular matrix that impedes drug penetration.

Clotrimazole is one of the antifungal agents commonly used for the treatment of VVC. In the current study, reduced susceptibility to clotrimazole was observed in 11.4% of the isolates tested, including eight *C. albicans* isolates and one *C. parapsilosis* isolate. Resistance rates to clotrimazole vary significantly between countries and these variations in resistance may be related to differences in epidemiological surveillance resources, the methods used for in vitro susceptibility testing, and prolonged exposure to this antifungal agent, particularly among women with RVVC [34,36,38,40,64].

In the current study, reduced susceptibility to miconazole was observed in 47.1% of the fungal isolates tested. The proportion of *C. albicans* isolates with reduced susceptibility was significantly higher among patients with recurrent vulvovaginal candidiasis (58.4%) compared to those with acute infection (44.8%). Variable resistance rates to miconazole have been reported linked to the widespread use of miconazole as an OTC treatment for VVC. In many cases, excessive or inappropriate use of this antifungal agent has been associated with recurrent episodes caused by *Candida* species with reduced susceptibility to miconazole [29,34,37,38,40,62,65].

Fluconazole is arguably the most widely used antifungal agent for the treatment of both AVVC and RVVC, although its efficacy has declined over time [28]. In the current study, reduced in vitro susceptibility to fluconazole was observed in 24.3% of *C. albicans* isolates. However, a study by Orellana Quito et al. [16] in Azuay, Ecuador, reported a lower resistance rate of 14.3%. Several studies have shown resistance rates comparable to those found in our study [31,34,36,38,40,65,66]. No significant differences were found between isolates from patients with AVVC and RVVC. However, a study conducted in Ghana reported higher fluconazole resistance among isolates from patients with RVVC compared to those with acute infection [66]. Additionally, it has been reported that approximately half of the patients treated with fluconazole experience recurrence within six months, likely due to frequent and/or prolonged use of the drug [3,4,48].

In the present study, non-*C. albicans* species did not show reduced susceptibility to fluconazole. However, a systematic review of VVC in several Ibero-American countries reported reduced azole susceptibility rates, specifically to fluconazole, ranging from 4% to 100% among *C. glabrata* isolates [46].

All *Candida* isolates in the current study, regardless of species or genotype, were susceptible to itraconazole, consistent with the low itraconazole resistance reported in previous studies [40,43,67]. However, other studies [34,36,67] have reported resistance to itraconazole among various *Candida* species. Liu et al. [67] found that genotype B isolates had the highest resistance rate (66.7%), while genotypes A and C were fully susceptible. Although the number of *Candida albicans* genotype B isolates was limited in our study, in vitro susceptibility to itraconazole was comparable to that observed in other *C. albicans* isolates.

It has been difficult to compare the findings of the current study with those of other studies conducted in Ecuador. This is partly due to the limited number of studies on the prevalence of VVC, the distribution of the disease by clinical presentation or age, the frequency of causative species, or antifungal resistance patterns. The regional differences in resistance patterns observed in our study compared to those reported by other Ecuadorian studies highlight the importance of evaluating the specific circumstances of each medical centre, patient group, and individual patient in order to establish a dynamic epidemiological profile that enables the most appropriate treatment.

In conclusion, this study provides valuable insights into the epidemiology of VVC in Ecuador. We assessed disease prevalence, identified risk factors, and the responsible aetiological agents, along with their in vitro susceptibility to polyenes and azoles used in both topical and systemic treatment. Ongoing surveillance is essential to understand the changes in resistance patterns and for developing effective treatment protocols. The current data can guide clinicians in managing VVC in Ecuador, incorporating resistance trends in treatment decisions and in considering alternative therapies.

## Figures and Tables

**Table 1 jof-11-00742-t001:** Demographic characteristics and predisposing factors in patients with acute (AVVC) and recurrent vulvovaginal candidiasis (RVVC) attending consultations at the Medical Service of PUCE.

Demographic Data	Patients			
	Non VVC	AVVC	RVVC	Total
N (%)	124 (63.6)	56 (28.7)	15 (7.7)	195 (100)
Age range (years)	18–58	18–67	21–45	18–67
Mean age (years)	24.5	25.8	30.5	25.3
Antibiotics [n (%)]	11 (9.9)	4 (7.1)	1 (6.7)	16 (8.2)
Contraceptives [n (%)]	82 (66.1)	27 (48.2)	7 (46.7)	116 (59.5)
Unique sexual partner [n (%)]	98 (79)	18 (32.1)	8 (53.3)	124 (63.6)
Two or more sexual partners [n (%)]	26 (20.9)	38 (67.9)	7 (46.7)	71 (36.4)
Without predisposing factors [n (%)]	31 (25)	24 (42.9)	7 (46.7)	62 (31.8)

**Table 2 jof-11-00742-t002:** Fungal species isolated from patients with acute and recurrent vulvovaginal candidiasis.

Acute vulvovaginal candidiasis (N = 56)			
Fungal species	Pure culture N (%)	Mixed culture N (%)	Species in mixed culture (N)
*Candida albicans*	45 (80.4) *	7 (12.5)	*Saccharomyces cerevisiae* (3) *Candida glabrata* (2) *Candida famata* (2)
*Candida glabrata* (*Nakaseomyces glabratus*)	2 (3.6)	2 (3.6)	*Candida albicans* (2)
*Saccharomyces cerevisiae*	1 (1.8)	3 (5.4)	*Candida albicans* (3)
*Candida parapsilosis*	1 (1.8)	0	
*Candida famata* (*Debariomyces hansenii*)	0	2 (3.6)	*Candida albicans* (2)
Recurrent vulvovaginal candidiasis (N = 15)			
Fungal species	Pure culture N (%)	Mixed culture N (%)	Species in mixed culture
*Candida albicans*	9 (60)	3 (20)	*Saccharomyces cerevisiae* (2) *Candida glabrata* (1)
*Saccharomyces cerevisiae*	2 (13.3)	2 (13.3)	*Candida albicans* (2)
*Candida glabrata* (*Nakaseomyces glabratus*)	1 (6.7)	1 (6.7)	*Candida albicans* (1)
All vulvovaginal candidiasis (N = 71)			
Fungal species	Pure culture N (%)	Mixed culture N (%)	Species in mixed culture
*Candida albicans*	54 (76.1) *	10 (14.1)	*Saccharomyces cerevisiae* (5) *Candida glabrata* (3) *Candida famata* (2)
*Candida glabrata* (*Nakaseomyces glabratus*)	3 (4.2)	3 (4.2)	*Candida albicans* (3)
*Saccharomyces cerevisiae*	3 (4.2)	5 (7)	*Candida albicans* (5)
*Candida parapsilosis*	1 (1.4)	0	
*Candida famata* (*Debariomyces hansenii*)	0	2 (2.8)	*Candida albicans* (2)

* In six vaginal specimens from women suffering from AVVC, two genotypes of *Candida albicans* were isolated, exhibiting colonies with slightly different morphologies.

**Table 3 jof-11-00742-t003:** Susceptibility profiles of *Candida albicans* isolates from patients with acute (AVVC) and recurrent vulvovaginal candidiasis (RVVC). Isolates were classified as susceptible (S), susceptible dose-dependent/intermediate (SDD/I), or resistant (R) to the antifungal agents evaluated.

Antifungal Drug		No. of Isolates (% of Isolates) in Each Category		
Nystatin (50 µg/mL)		S (≥15 mm)	I (10–14 mm)	R (≤9 mm)
	AVVC	58 (100)	0	0
	RVVC	12 (100)	0	0
	Total	70 (100)	0	0
Amphotericin B		S (≤1 µg/mL)		R (≥2 µg/mL)
	AVVC	57 (98.3)		1 (1.7)
	RVVC	11 (91.7)		1 (8.3)
	Total	68 (97.1)		2 (2.9)
Clotrimazole (50 µg/mL)		S (≥20 mm)	I (12–19 mm)	R (≤11 mm)
	AVVC	54 (93.1)	3 (5.2)	1 (1.7)
	RVVC	8 (66.7)	3 (25)	1 (8.3)
	Total	62 (88.6)	6 (8.6)	2 (2.9)
Miconazole (50 µg/mL)		S (≥20 mm)	I (12–19 mm)	R (≤11 mm)
	AVVC	32 (55.2)	2 (3.4)	24 (41.4)
	RVVC	5 (41.6)	2 (16.7)	5 (41.7)
	Total	37 (52.9)	4 (5.7)	29 (41.4)
Fluconazole		S (≤1 µg/mL)	SDD (4 µg/mL)	R (≥8 µg/mL)
	AVVC	44 (75.9)	9 (15.5)	5 (8.6)
	RVVC	9 (75)	2 (16.7)	1 (8.3)
	Total	53 (75.7)	11 (15.7)	6 (8.6)
Itraconazole		S (≤0.125 µg/mL)	SDD (0.25–0.5 µg/mL)	R (≥1 µg/mL)
	AVVC	58 (100)	0	0
	RVVC	12 (100)	0	0
	Total	70 (100)	0	0

**Table 4 jof-11-00742-t004:** Antifungal activity of amphotericin B, fluconazole and itraconazole against *Candida albicans* isolates from acute vulvovaginal candidiasis (AVVC) and recurrent vulvovaginal candidiasis (RVVC) patients.

Antifungal Drug		No. of Isolates at MIC (µg/mL) (Cumulative %)											
		≤0.03	0.06	0.125	0.25	0.5	1	2	4	8	16	32	≥64
Amphotericin B													
	AVVC	1 (1.6)	13 (21)	52 (83.9)	57 (92)	59 (95.2)	60 (96.8)	60 (96.8)	62 (100)				
	RVVC	1 (12.5)	1 (12.5)	7 (87.5)	8 (100)								
	Total	2 (2.9)	14 (20)	59 (84.2)	65 (92.9)	67 (95.7)	68 (97.1)	68 (97.1)	70 (100)				
Fluconazole													
	AVVC				32 (54.2)	35 (59.3)	37 (62.7)	43 (72.9)	44 (74.7)	56 (91.5)	59 (96.6)	62 (100)	
	RVVC				3 (36.4)	5 (63.7)	5 (63.7)	6 (81.9)	6 (81.9)	7 (91.1)	8 (100)		
	Total				35 (50)	40 (57.1)	42 (60)	49 (70)	50 (71.4)	63 (90)	67 (95.7)	70 (100)	
Itraconazole													
	AVVC	35 (56.9)	45 (72.4)	53 (86.2)	60 (98.3)	62 (100)							
	RVVC	2 (25)	3 (37.5)	3 (37.5)	5 (62.5)	5 (62.5)	7 (87.5)	8 (100)					
	Total	37 (50)	48 (57.1)	56 (60)	65 (70)	67 (71.4)	69 (90)	70 (100)					

## Data Availability

The original contributions presented in this study are included in the article. Further inquiries can be directed to the corresponding author.

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
