# Peer review of "Aetiology of Vulvovaginal Candidiasis in Ecuador and In Vitro Antifungal Activity Against *Candida* Vaginal Isolates"

_jof, 2025, doi:10.3390/jof11100742_

Round 1
Reviewer 1 Report
The manuscript deals with epidemiological data on vulvovaginal candidiasis in Ecuador associated with susceptibility data. The article is well constructed and provides important insights into the difficulties of treating and identifying the etiological agent.
The chromogenic medium used also identifies C. tropicalis, and although the authors did not find any cases, I think it is pertinent that this information be included in the materials and methods. Another point is that the species Candida glabrata can have colors other than creamy white; it can acquire colors such as mauve (https://www.chromagar.com/en/product/chromagar-candida/).
Amphotericin B is not used as a treatment protocol for VVC. I understand the importance of measuring its susceptibility, but I believe this should be explained in the results.
Why do the authors believe that nystatin and itraconazole were the best therapeutic options in the susceptibility tests?
The authors mention that C. krusei exhibits lower susceptibility to azoles, when in fact the species is intrinsically resistant to fluconazole. I suggest revising this wording.
There is a layout error in Table 1 (row 4, column 2).
It is not clear to me why the authors used two susceptibility methods. Wouldn't broth microdilution be superior to disk diffusion for quantitative results?

Author Response
First and foremost, we would like to express our sincere gratitude to this reviewer for their valuable comments, corrections, and suggestions. We genuinely believe that the overall quality of the manuscript has been significantly enhanced.
Major comments
The manuscript deals with epidemiological data on vulvovaginal candidiasis in Ecuador associated with susceptibility data. The article is well constructed and provides important insights into the difficulties of treating and identifying the etiological agent.
Detailed comments
Q1. The chromogenic medium used also identifies Candida tropicalis, and although the authors did not find any cases, I think it is pertinent that this information be included in the materials and methods. Another point is that the species Candida glabrata can have colors other than creamy white; it can acquire colors such as mauve (https://www.chromagar.com/en/product/chromagar-candida/).
In accordance with this suggestion, we have revised the text to include mauve as one of the possible colours of Candida glabrata colonies, and blue as the colour observed in Candida tropicalis colonies (see lines 111 and 112).
Q2. Amphotericin B is not used as a treatment protocol for VVC. I understand the importance of measuring its susceptibility, but I believe this should be explained in the results.
In response to this important suggestion, we have included a comment highlighting the relevance of performing in vitro susceptibility testing for amphotericin B, considering that it is one of the gold standards in antifungal activity studies.
We have included the following sentence in lines 177 and 178: Amphotericin B was included as the reference comparator (gold standard) for the in vitro antifungal activity of the other drugs evaluated in this study.
Q3. Why do the authors believe that nystatin and itraconazole were the best therapeutic options in the susceptibility tests?
In our countries (Ecuador and Spain), nystatin (vaginal tablets) and itraconazole (oral capsules) are considered alternative options in clinical protocols when therapeutic failure occurs with topical azole treatments or oral fluconazole.
Q4. The authors mention that C. krusei exhibits lower susceptibility to azoles, when in fact the species is intrinsically resistant to fluconazole. I suggest revising this wording.
This comment is absolutely true, and we have revised the sentence to clearly state the intrinsic resistance of Candida krusei to fluconazole.
Q5. There is a layout error in Table 1 (row 4, column 2).
We have corrected this layout error.
Q6. It is not clear to me why the authors used two susceptibility methods. Wouldn't broth microdilution be superior to disk diffusion for quantitative results?
You are right. In vitro susceptibility testing using broth dilution is superior to disc diffusion for obtaining quantitative results. However, CLSI methods are not standardised for miconazole and nystatin, and for practical reasons, agar diffusion using commercially available tablets of these two antifungal agents was employed.
Reviewer 2 Report
The manuscript presents a timely epidemiological study on vulvovaginitis caused by Candida species in Ecuadorian patients. I share the authors' opinion about the lack of updated information on this subject in this country. Although the results are of local relevance, I anticipate they will be welcomed by international specialists.
I offer the following points for manuscript strength:
In the exclusion criteria, it is mentioned pregnant patients, but a few lines below, the authors indicate that they included pregnant patients in the study, please modify accordingly.
There is no mention of whether the authors found bacteria and fungal coinfections.
It would be relevant to include the treatment used for infections and the outcome. A possible correlation between antifungal drug susceptibility and the drugs working in the clinical setting may be generated if the information is available.
The manuscript presents a timely epidemiological study on vulvovaginitis caused by Candida species in Ecuadorian patients. I share the authors' opinion about the lack of updated information on this subject in this country. Although the results are of local relevance, I anticipate they will be welcomed by international specialists.
I offer the following points for manuscript strength:
In the exclusion criteria, it is mentioned pregnant patients, but a few lines below, the authors indicate that they included pregnant patients in the study, please modify accordingly.
There is no mention of whether the authors found bacteria and fungal coinfections.
It would be relevant to include the treatment used for infections and the outcome. A possible correlation between antifungal drug susceptibility and the drugs working in the clinical setting may be generated if the information is available.
Author Response
First and foremost, we would like to express our sincere gratitude to this reviewer for their valuable comments, corrections, and suggestions. We genuinely believe that the overall quality of the manuscript has been significantly enhanced.
Major comments
The manuscript presents a timely epidemiological study on vulvovaginitis caused by Candida species in Ecuadorian patients. I share the authors' opinion about the lack of updated information on this subject in this country. Although the results are of local relevance, I anticipate they will be welcomed by international specialists.
I offer the following points for manuscript strength:
In the exclusion criteria, it is mentioned pregnant patients, but a few lines below, the authors indicate that they included pregnant patients in the study, please modify accordingly.
There is no mention of whether the authors found bacterial and fungal coinfections.
It would be relevant to include the treatment used for infections and the outcome. A possible correlation between antifungal drug susceptibility and the drugs working in the clinical setting may be generated if the information is available.
Detailed comments
The manuscript presents a timely epidemiological study on vulvovaginitis caused by Candida species in Ecuadorian patients. I share the authors' opinion about the lack of updated information on this subject in this country. Although the results are of local relevance, I anticipate they will be welcomed by international specialists.
I offer the following points for manuscript strength:
Q1. In the exclusion criteria, it is mentioned pregnant patients, but a few lines below, the authors indicate that they included pregnant patients in the study, please modify accordingly.
In accordance with this comment, this error has been corrected in lines 93 and 94.
Q2. There is no mention of whether the authors found bacterial and fungal coinfections.
We have revised the text to incorporate this information. It is now described as follows:
Lines 232-234: Nine of the 51 patients with AVVC had mixed infections with Gardnerella vaginalis (17.6%), and one patient had a mixed infection with Trichomonas vaginalis (1.9%).
Lines 236-238: Their symptoms were attributed to other vaginal conditions, including bacterial vaginosis (27 patients, 21.8%), trichomoniasis (37 patients, 29.8%), both conditions concurrently (6 patients, 4.8%) or to non-infectious genital conditions (54 patients, 43.6%).
Q3. It would be relevant to include the treatment used for infections and the outcome. A possible correlation between antifungal drug susceptibility and the drugs working in the clinical setting may be generated if the information is available.
Following the reviewer’s recommendations, we have included in the manuscript the treatments administered to patients, which were aligned with internationally approved therapeutic guidelines. However, due to low patient adherence, it was nearly impossible to evaluate treatment outcomes. As this is not a routine practice, fewer than 50% of patients returned for follow-up after treatment. It could be assumed that these patients were cured and that the antifungal treatment was effective. Nevertheless, it is also likely that some patients did not experience symptom improvement and sought care from other professionals, used over-the-counter (OTC) products, or resorted to alternative remedies. Moreover, according to standard protocols, the use of oral fluconazole in combination with a topical azole or nystatin complicates the correlation between clinical cure and in vitro antifungal susceptibility in those patients who did return for follow-up after receiving treatment.
The text included in the manuscript in lines 510–520 is as follows:
The therapeutic protocol followed at Medical Service of the PUCE, based on inter-national guidelines recommendations [4,21], was as follows: For patients with AVVC, oral treatment with 150 mg fluconazole was administered as an initial dose, followed by a second dose after 72 hours. This was accompanied by topical treatment with vaginal ovules containing either 200 mg clotrimazole or 400 mg miconazole for seven days. As an alternative to these topical azoles, a daily vaginal tablet of nystatin for seven days was used in complicated AVVC. For patients with RVVC, a suppression regimen was implemented with oral fluconazole 150 mg on days 1, 4, and 7, followed by weekly maintenance therapy for six months. During this maintenance period, patients also received a weekly vaginal ovule containing either 500 mg clotrimazole or 400 mg miconazole. As an alternative to these topical azoles, a weekly vaginal tablet of nystatin was also used.
Reviewer 3 Report
In the manuscript "Aetiology of vulvovaginal candidiasis in Ecuador and in vitro antifungal activity against Candida vaginal isolates" the authors conducted a 12-month prospective study to evaluate the etiology and antifungal resistance patterns in Ecuadorian patients with vulvovaginal candidiasis. This is an interesting study that provides valuable epidemiological information, considering that this mycosis is one of the most common in the world. Overall, it is a well-structured study; however, I have some comments.
Materials and methods
On lines 117 and 118, the authors mention "Species identification was confirmed by PCR using species-specific primers [18-21]." However, these references do not refer to the identification of the isolates by PCR. It is very important to correct the references and describe the methodology used for the molecular identification of the different Candida species.
In section “2.3. Genotyping of Candida albicans isolate” lines 141-145, the authors mention “PCR products were resolved by electrophoresis for 70 minutes at 90 V in a horizontal 141 Sub-cell GT chamber (Bio-Rad Laboratories, California, USA) using 1.5% low 142 J. Fungi 2025, 11, x FOR PEER REVIEW 4 of 17 electroendosmosis (EEO) agarose gel stained with ethidium bromide. A Hyperladder IV 143 molecular weight marker (Bioline), comprising ten bands ranging from 100 to 1000 base 144 pairs, was used to estimate fragment sizes. Gel visualization was performed using the Gel 145 Chemidoc imaging system (Bio-Rad)” however they do not mention how the genotypes were assigned to each Candida albicans isolate, by the size and number of the bands obtained for each isolate? Therefore, I suggest you clarify this section.
Minor comments
I suggest the authors review the entire manuscript and correct the species names. The first time a species appears in the text, it should be spelled out in full, for example, Candida albicans. Subsequent occurrences should simply be abbreviated, for example, C. albicans.
none
Author Response
First and foremost, we would like to express our sincere gratitude to this reviewer for their valuable comments, corrections, and suggestions. We genuinely believe that the overall quality of the manuscript has been significantly enhanced.
Major comments
In the manuscript "Aetiology of vulvovaginal candidiasis in Ecuador and in vitro antifungal activity against Candida vaginal isolates" the authors conducted a 12-month prospective study to evaluate the etiology and antifungal resistance patterns in Ecuadorian patients with vulvovaginal candidiasis. This is an interesting study that provides valuable epidemiological information, considering that this mycosis is one of the most common in the world. Overall, it is a well-structured study; however, I have some comments.
Materials and methods
Q1. On lines 117 and 118, the authors mention "Species identification was confirmed by PCR using species-specific primers [18-21]." However, these references do not refer to the identification of the isolates by PCR. It is very important to correct the references and describe the methodology used for the molecular identification of the different Candida species.
To clarify, these references have been removed from the original paragraph and relocated. We have reviewed the molecular techniques employed in this study and incorporated two pertinent references: Romeo & Criseo (2008) [21], concerning the molecular identification of the Candida albicans complex, and Romeo et al. (2009) [22], regarding the molecular identification of the Candida glabrata complex. The reference [23], pertaining to the molecular identification of the Candida parapsilosis complex, has been retained (lines 123-131).
Q2. In section “2.3. Genotyping of Candida albicans isolate” lines 141-145, the authors mention “PCR products were resolved by electrophoresis for 70 minutes at 90 V in a horizontal 141 Sub-cell GT chamber (Bio-Rad Laboratories, California, USA) using 1.5% low 142 J. Fungi 2025, 11, x FOR PEER REVIEW 4 of 17 electroendosmosis (EEO) agarose gel stained with ethidium bromide. A Hyperladder IV 143 molecular weight marker (Bioline), comprising ten bands ranging from 100 to 1000 base 144 pairs, was used to estimate fragment sizes. Gel visualization was performed using the Gel 145 Chemidoc imaging system (Bio-Rad)” however they do not mention how the genotypes were assigned to each Candida albicans isolate, by the size and number of the bands obtained for each isolate? Therefore, I suggest you clarify this section.
To clarify this section, we have added further information regarding Candida albicans genotypes and their corresponding PCR fragment sizes at lines 159-161. The relevant reference has also been included in both the main text and the reference list. The added text read as follows: “Based on PCR fragment sizes, Candida albicans genotypes were classified as follows: genotype A, 450 bp; genotype B, 840 bp; genotype C, 450 and 840 bp; genotype D, 1080 bp; and genotype E, 1400 bp [24]”.
Minor comments
Q3. I suggest the authors review the entire manuscript and correct the species names. The first time a species appears in the text, it should be spelled out in full, for example, Candida albicans. Subsequent occurrences should simply be abbreviated, for example, C. albicans.
We have taken this suggestion into account and made the appropriate amendments to the text
Reviewer 4 Report
The study "Aetiology of vulvovaginal candidiasis in Ecuador and in vitro antifungal activity against Candida vaginal isolates" by Bowen et al., is an epidemiological investigation on VVC which attempted to draw a picture on prevalence, risk factors, and the Candida species involved and their antifungal profiles, in Ecuador.
The research is very well designed, conducted, and written. It's a great addition to the knowledge on the state of VVC in Ecuador. I couldn't find any major critic, just minor revisions below:
Line 19 in Abstract should read "AVVC" instead of the typo ACCV
Line 89 to 93: it is unclear why only non-C. albicans complicated VVC cases were included?
Line 118: references 18 & 19, do not mention or set-up any PCR or species-specific primers for Candida species identification. Should be removed. Normally, universal fungal primers should be used for PCR, sequencing, then sequence should be blasted against NCBI sequence database to positively identify species, usually with >= 99 nucleotide identity. Please describe with more details the procedure: did you use a single colony? How mixed species infections were initially detected?
Line 126: It could be interesting to show examples of genotyping results (may be even in Suppl. material). Is there a reference that could be cited for this technique?
Line 257: not sure how 70 C. albicans isolates were genotyped, while the table-2 describes a total of 64 C. albicans isolates? Where the balance of 6 isolates came from?
Line 433: from 0.2% to 2% in surveillance studies "in Malaysia"? (indicate where study was conducted)
Line 542: other Ecuadorian "studies" will sound better than "authors"
Author Response
First and foremost, we would like to express our sincere gratitude to this reviewer for their valuable comments, corrections, and suggestions. We genuinely believe that the overall quality of the manuscript has been significantly enhanced.
Major comments
The study "Aetiology of vulvovaginal candidiasis in Ecuador and in vitro antifungal activity against Candida vaginal isolates" by Bowen et al., is an epidemiological investigation on VVC which attempted to draw a picture on prevalence, risk factors, and the Candida species involved and their antifungal profiles, in Ecuador.
The research is very well designed, conducted, and written. It's a great addition to the knowledge on the state of VVC in Ecuador. I couldn't find any major critic, just minor revisions below:
Detailed comments
Q1. Line 19 in Abstract should read "AVVC" instead of the typo ACCV
Following this suggestion, we have corrected this mistake in the manuscript text.
Q2. Line 89 to 93: it is unclear why only non-C. albicans complicated VVC cases were included?
Given the characteristics of the patients attending the medical services at PUCE, with an almost negligible number of pregnant women, or those suffering from immunodeficiencies, diabetes, and other endocrinological disorders, it was considered during the study design that their inclusion could distort the clinical groups of patients presenting vaginal symptoms and fail to reflect the true epidemiological profile of this condition.
Q3. Line 118: references 18 & 19, do not mention or set-up any PCR or species-specific primers for Candida species identification. Should be removed. Normally, universal fungal primers should be used for PCR, sequencing, then sequence should be blasted against NCBI sequence database to positively identify species, usually with >= 99 nucleotide identity. Please describe with more details the procedure: did you use a single colony? How mixed species infections were initially detected?
To clarify, these references have been removed from the original paragraph and relocated. We have reviewed the molecular techniques employed in this study and added two relevant references in the references section: Romeo & Criseo (2008) [21], for the molecular identification of the Candida albicans complex, and Romeo et al. (2009) [22], for the molecular identification of the Candida glabrata complex. The reference [23], concerning the molecular identification of the Candida parapsilosis complex, has been retained.
Q4. Line 126: It could be interesting to show examples of genotyping results (may be even in Suppl. material). Is there a reference that could be cited for this technique?
To clarify this section, we have added further information regarding Candida albicans genotypes and their corresponding PCR fragment sizes at lines 159-161. The relevant reference has also been included in both the main text and the reference list. The added text read as follows: “Based on PCR fragment sizes, Candida albicans genotypes were classified as follows: genotype A, 450 bp; genotype B, 840 bp; genotype C, 450 and 840 bp; genotype D, 1080 bp; and genotype E, 1400 bp [23]”.
Q5. Line 257: not sure how 70 C. albicans isolates were genotyped, while the table-2 describes a total of 64 C. albicans isolates? Where the balance of 6 isolates came from?
This numerical imbalance is explained by the fact that in six vaginal specimens from women suffering from AVVC, two genotypes of Candida albicans were isolated, exhibiting colonies with slightly different morphologies. This has been addressed by including an explanatory note in the table footnote.
Q6. Line 433: from 0.2% to 2% in surveillance studies "in Malaysia"? (indicate where study was conducted)
The study of Kam and Xu was conducted in the area of Toronto-Hamilton, Ontario, Canada.
Q7. Line 542: other Ecuadorian "studies" will sound better than "authors" Following this suggestion, we have implemented this modification in the manuscript text.
Round 2
Reviewer 3 Report
In the new version of the manuscript "Aetiology of vulvovaginal candidiasis in Ecuador and in vitro antifungal activity against Candida vaginal isolates," the authors made changes based on the comments received to improve the work. However, I still have some concerns, particularly regarding the molecular identification of Candida species.
In the previous version, I requested that the references related to "Species identification was confirmed by PCR using species-specific primers [18-21]" be corrected and that the methodology used for the molecular identification of the different Candida species be described. However, although the references were corrected, in this new version, it is not clear how many isolates were identified by conventional methods through serum germ tube production, chlamydoconidia formation, and biochemical profiling? Of these, how many isolates were identified with each of the methodologies mentioned? And what was the justification for using three methodologies instead of just one, for example, the use of ITS1 and ITS4 markers for their identification?
In the discussion, they don't mention the results obtained related to molecular identification. What advantages does each of the methods used have? Is molecular identification important? I suggest you discuss this.
Author Response
Q1. In the previous version, I requested that the references related to "Species identification was confirmed by PCR using species-specific primers [18-21]" be corrected and that the methodology used for the molecular identification of the different Candida species be described. However, although the references were corrected, in this new version, it is not clear how many isolates were identified by conventional methods through serum germ tube production, chlamydoconidia formation, and biochemical profiling? Of these, how many isolates were identified with each of the methodologies mentioned? And what was the justification for using three methodologies instead of just one, for example, the use of ITS1 and ITS4 markers for their identification?
In the discussion, they don't mention the results obtained related to molecular identification. What advantages does each of the methods used have? Is molecular identification important? I suggest you discuss this.
Thank you very much for your insightful comment.
In accordance with our laboratory’s standard protocol for yeast identification, all vaginal isolates underwent germ tube formation testing, chlamydospore production on corn meal agar, biochemical profiling using ATB ID 32C (bioMérièux), and assessment of colony morphology on CHROMagar Candida (and occasionally on CHROMID Candida).
Additionally, for 14 Candida albicans isolates with incomplete concordance across these tests, molecular identification was performed to confirm whether they were Candida albicans or another species within the Candida albicans complex (i.e. Candida dubliniensis or Candida africana). Following this protocol, molecular confirmation was also carried out for all Candida glabrata isolates and the single Candida parapsilosis isolate, to rule out the presence of Candida bracarensis, Candida nivariensis, Candida metapsilosis, or Candida orthopsilosis, or to confirm identification as Candida glabrata sensu stricto or Candida parapsilosis sensu stricto. These procedures are routinely performed in our laboratory due to the relatively frequent isolation of these cryptic species within the Candida albicans, Candida glabrata, and Candida parapsilosis complexes.
We had not considered it necessary to include this explanation in the Discussion section. However, following the reviewer’s valuable suggestion, we have now added the following statement (Lines 424-427):
“Molecular confirmation of vaginal Candida isolates was essential to exclude cryptic species within the Candida albicans, Candida glabrata, and Candida parapsilosis complexes, which may exhibit distinct pathogenic traits and in vitro antifungal susceptibility profiles.”
Round 3
Reviewer 3 Report
I thank the authors for their attention, and after reviewing the new version of the manuscript “Aetiology of vulvovaginal candidiasis in Ecuador and in vitro antifungal activity against Candida vaginal isolates” I agree that it be published.
None